# Structural Equation Modeling (SEM): Gaming Disorder Leading Untreated Attention-Deficit/Hyperactivity Disorder to Disruptive Mood Dysregulation

**DOI:** 10.3390/ijerph19116648

**Published:** 2022-05-29

**Authors:** Ruu-Fen Tzang, Chuan-Hsin Chang, Yue-Cune Chang

**Affiliations:** 1Department of Psychiatry, Mackay Memorial Hospital, Taipei 10449, Taiwan; rf.tzang@gmail.com; 2Department of Childhood Care and Education, Mackay Junior College of Medicine, Nursing and Management, Taipei 11260, Taiwan; 3Department of Audiology and Speech-Language Pathology, Mackay Medical College, New Taipei City 25245, Taiwan; 4Agricultural Biotechnology Research Center, Academia Sinica, Taipei 11529, Taiwan; chuanhsin032484@gmail.com; 5Department of Mathematics, Tamkang University, Taipei 251301, Taiwan

**Keywords:** ADHD, IGD, DMDD, mediator, SEM

## Abstract

(1) Background: Internet gaming disorder (IGD) in youths likely leads to disruptive mood dysregulation, especially among those with attention-deficit/hyperactivity disorder (ADHD). Whether IGD mediates the pathways leading ADHD to disruptive emotional dysfunction remains unclear. This study aims to elucidate the direct or indirect influence of IGD on ADHD; (2) Method: The Swanson, Nolan, and Pelham Version IV questionnaire was used to evaluate symptoms of ADHD and oppositional defiant disorder, and the Chen gaming disorder scale was used to measure IGD. A psychiatrist diagnosed ADHD, IGD, and disruptive mood dysregulation disorder (DMDD)-like symptoms. Structural equation modeling was applied to evaluate the role of IGD in mediating ADHD progression to disruptive mood dysregulation; (3) Results: Among a total of 102 ADHD youths, 53 (52%) of them with IGD were significantly more likely to have poor interpersonal relationships (*p* < 0.01) and DMDD-like symptoms (*p* < 0.01) than ADHD youths without IGD. IGD played a mediating role in increasing the risk of disruptive mood dysregulation in ADHD youths; (4) Conclusions: The findings suggest that IGD mediates ADHD’s progression to disruptive mood dysregulation. Intensive biopsychosocial interventions are warranted for ADHD youths with IGD. More children and adolescents became mood-dysregulated after excessive gaming during the COVID-19 pandemic; this study’s results suggest that child mental health experts develop earlier detection and prevention strategies for children and adolescents hidden behind internet addiction.

## 1. Introduction

Internet gaming disorder (IGD), a new mental disorder, has become prevalent among children and adolescents recently, especially during the outbreak of the coronavirus disease (COVID-19). Under the lockdown crisis, many adolescents increasingly engaged in internet gaming [1,2], thereby raising the rate of children and adolescents with internet gaming disorder [3,4].

There is some overlap between children and adolescents with pathological internet gaming disorder compared to internet addiction. Following a developmentally oriented approach, researchers recently indeed found that internet addiction among children and adolescents is closely correlated with their specific emotional–psychopathological characteristics [5]. Those children with pre-existing mental disorders, such as attention-deficit/hyperactivity disorder (ADHD), tended to excessively use the internet during the COVID-19 pandemic [6] and consequently worsen ADHD severity, emotional dysregulation [7], and temper outbursts [8]. Children with ADHD were more likely to develop the mental disorder called disruptive mood dysregulation disorder (DMDD), characterized by long-term dysphoria with at least three severe anger episodes per week for a year [9,10].

IGD was reported to play a mediating role in leading children with ADHD to escape from tedious learning processes [11]. ADHD and IGD were commonly observed and closely correlated mental disorders _ENREF_5 [12]. Approximately 83% of youths with IGD had ADHD [13]. Further, the severity of IGD symptoms was also related to ADHD severity [14]. Noteworthily, youths with ADHD became more aggressive, violent, or delinquent when they spent more time watching highly dangerous content in internet gaming sessions [15].

IGD had high prevalence rates among children and adolescents, ranging from 4–6% (in European countries) to 13.5% (in China) [16]. It caused not only physical adverse effects (such as neck muscle soreness and early cataract) but also more psychological consequences, including negative psychologic wellbeing [17], school refusal and social withdrawal or so-called hikikomori syndrome [18,19], internet-related cognitive bias and coping [20], anxiety, depression, and impaired social and family life [21]. Recent research found that unbridled internet usage in early adolescence was closely related to impulsive behavior [22,23], temper loss [24], or disruptive behavior disorder [25]. As a result, young pathological internet gamers had a higher risk of impulsive, aggressive, and violent behaviors [26]. Children with ADHD and oppositional defiant disorder (ODD) had more severe mood dysregulation [27].

In summary, ADHD is a neurodevelopmental disorder commonly noticed among children and adolescents, with prevalence of 9%. IGD is a new formal mental disorder and is also very commonly seen recently among children and adolescents with prevalence ranging from 4–6% (in European countries) to 13.5% (in China). This study result highlights that if children and adolescents with untreated ADHD are internet-addicted, they will experience a mood-disrupted state instead of depression. Depression is a mood disorder commonly noticed among adults instead of children and adolescents. Here, we suggest that people of children’s mental health expertise face this recent surge in youth with IGD properly by providing early preventive intervention in IGD to prevent untreated ADHD from becoming DMDD, which brings about negative influences on their personality development.

IGD possibly mediated ADHD youth’s development of DMDD-like symptoms and more psychiatric disorders, such as more severe ADHD, mood disorders, self-injury, eating disorders, ODD, conduct disorder (CD), personality disorders, and substance use disorders during the critical period of the COVID-19 pandemic [28]. For the early prevention of more children and adolescents becoming mood-dysregulated after excessive gaming during the COVID-19 pandemic, this study result suggests that child mental health experts develop earlier detection and prevention strategies for children and adolescents hidden behind internet addiction.

How IGD leads children with ADHD to become irascible and display DMDD-like symptoms deserves study. We hypothesized that IGD may play a mediating role in leading children with ADHD to develop DMDD-like symptoms and tested this hypothesis using structural equation modeling (SEM). To our knowledge, this is the first study to examine whether DMDD may be a consequence for ADHD adolescents with gaming disorders. The results may help mental health experts to develop an early detection and prevention strategy for ADHD, IGD, and DMDD among children and adolescents.

## 2. Materials and Methods

### 2.1. Participants

Patients were recruited from the outpatient units of Mackay Memorial Hospital (MMH), a major medical center in Taipei, Taiwan. The research protocol was approved by the MMH Institutional Review Board (IRB). Written informed consent was obtained in line with the IRB’s guidelines after complete description of the study to the children with ADHD and their parents. Subjects included children aged 7–18 years with a diagnosis of ADHD.

A child-and-adolescent psychiatrist confirmed the diagnoses of ADHD and other comorbid mental disorders using the criteria of the Diagnostic and Statistical Manual of Mental Health Disorders, 5th Edition (DSM-5) (American Psychiatric Association, 2013). Other comorbid psychiatric disorders included ODD, CD, unspecified anxiety disorder, unspecified depressive disorder, adjustment disorder, somatic symptom disorder, persistent (chronic) motor or vocal tic disorder, Tourette’s disorder, language disorder, and speech sound disorder.

In the DSM-5, IGD has been recognized as a condition for further research (American Psychiatric Association, 2013). While diagnosis of DMDD needs to fulfill the criteria of unreasonable mood dysregulation and the age at onset is before 10 years, we regarded our subjects as having DMDD-like symptoms because they had disruptive mood dysregulation but no history of mood dysregulation starting before 10 years old.

The exclusion criteria were as follows: pediatric patients or their parent(s)/caregiver(s) with known or suspected psychotic disorders, intellectual disabilities, or other severe mental conditions that would prevent them from completing the study.

This study’s recruitment was at the beginning of the COVID-19 situation in Taiwan. We performed a a special statistical analysis called structural equation modeling (SEM) applied to evaluate the role of IGD in mediating the development of disruptive mood dysregulation in children and adolescents with ADHD. We have no data detailing how severe the COVID-19 situation in Taiwan was during the study period to influence this study result.

### 2.2. Baseline Characteristics

Baseline characteristics of the ADHD children with or without IGD included: gender, school performance, interpersonal relationships, comorbidity (ODD, CD, unspecified anxiety disorder, unspecified depressive disorder, adjustment disorder, somatic symptom disorder, persistent motor or vocal tic disorder, Tourette’s disorder, language disorder, speech sound disorder, and DMDD-like symptoms), ADHD subtype, family psychiatric history, sibling suffering from ADHD, parent suffered from ADHD in childhood, the strategy of parents to deal with stress, parental understanding of ADHD, parental marital satisfaction, online chatting or playing games on working days ≥ 1 h, online chatting or playing games ≥ 3 h on holidays, drug response, parenting group therapy, compliance, age, height, weight, age of the father and mother, and the number of comorbidities.

### 2.3. Measures

Each subject recruited for this study was invited to participate in the following programs and was interviewed to derive the following measures.

#### 2.3.1. Chen Internet Addiction Scale (CIAS)

The CIAS is a self-reported questionnaire consisting of 26 questions on a four-point scale that assesses the five dimensions of internet use-related problems with good reliability and validity [29]. These dimensions are compulsive use, withdrawal, tolerance, interpersonal and health problems, and time-management problems. The CIAS exhibits good internal consistency of the scale, with Cronbach’s alpha values between 0.79 and 0.93 for the subscales. Higher CIAS scores indicate increased severity of gaming disorder. The CIAS also has good diagnostic accuracy (89.6%). The screening cut-off point in the original study had high sensitivity (85.6%), and the diagnostic cut-off point had the highest diagnostic accuracy, classifying 87.6% of the participants correctly.

#### 2.3.2. Swanson, Nolan, and Pelham Version IV Questionnaire (SNAP-IV)

The SNAP-IV is a widely used rating scale used to screen for ADHD. The SNAP-IV-26 screens for nine symptoms of ADHD’s hyperactive-impulsive type, nine symptoms of the inattentive ADHD type, and eight symptoms of oppositional defiant disorder as defined in the DSM-IV. The Chinese SNAP-IV demonstrated the satisfactory test–retest reliability (intraclass correlation = 0.59 to approximately 0.72) for the parent form. All subscales of both the parent and teacher forms displayed excellent internal consistency (alpha = 0.88 to approximately 0.90) [30].

#### 2.3.3. DMDD-Like Symptoms

DMDD is a new mental disease without any questionnaire or measurement scale at the time of this study. Therefore, we used a Likert scale, 0 to 3, to express the symptom severity of the DMDD criteria of the DSM-5.

### 2.4. Statistical Analysis

Structural equation modeling (SEM) was performed using AMOS software version 22.0 (maximum-likelihood method) to examine the direct or indirect relationships among ADHD, DMDD, and IA. The latent variable ADHD was indexed with three antecedent indicator variables: inattention, hyperactivity/impulsivity, and oppositional symptoms.

SEM was conducted to verify whether the proposed mediation model was suitable for the collected data. We used two models to estimate potential mediation effects: a basic model positing a direct relationship between ADHD and inattention, hyperactivity/impulsivity, and oppositional symptoms, and a mediation model positing a direct or indirect relationship among ADHD, DMDD, and IGD. The model fit indices were compared to recommend appropriate model fit indices in line with the effects of these factors. Model fits that represent how a SEM performance fits the sample data were assessed by five indices: the chi-square test (χ2), the goodness-of-fit index (GFI), the Tucker–Lewis Index (TLI), the comparative fit index (CFI), and the root-mean-squared error of approximation (RMSEA) [31]. The goodness-of-fit indicators (GFIs) were based on eight commonly used indices in SEM: the chi-square test (*p* > 0.05), standardized root-mean-square residual (SRMR) less than 0.05, root-mean-squared error of approximation (RMSEA) less than 0.06, GFI statistic greater than 0.95, incremental fit index (IFI) greater than 0.95, comparative fit index (CFI) greater than 0.95, normed fit index (NFI) greater than 0.95, and Tucker–Lewis index (TLI) greater than 0.95. The guidelines of these indices for determining model fitness were based on a previous study [32].

For SEM analysis, a minimum sample of 100 has been recommended by some experts [33]. Another good rule of thumb recommended by Bentler and Chou (1987) is to involve at least 15 participants for each observed variable [34]. Our sample size of 102 participants on this SEM analysis entails at least 15 participants for each observed variable and greatly exceeds the minimum requirements (15 × 5 = 75). Raw data were checked for normality and outliers prior to the analyses. List-wise deletion was used for 3 of the 105 participants with missing data on some of the variables at baseline because these omitted 3 cases had a lack of data on all variables at first.

## 3. Results

A total of 105 eligible ADHD children were enrolled, of whom 102 participants completed the baseline data of the three evaluation forms. The comparison of the baseline characteristics of ADHD with IA and non-IA groups is presented in Table 1. As anticipated, children with gaming disorders were more likely to have significantly bad interpersonal relationships than the non-addicted children (*p* = 0.008) group. Further, the gaming disorder group also had significantly higher comorbid diagnoses of DMDD and IGD than the non-addicted group (*p*-values = 0.006 and < 0.001, respectively). The zero-order correlations of the indicator variables are illustrated in Table 2.

The basic model depicted the direct relationship between ADHD and DMDD (Figure 1). The result of this model revealed that the (standardized) total direct effect of ADHD on DMDD was 0.62. In other words, due to the direct (unmediated) effect of ADHD on DMDD, when ADHD increased by 1 standard deviation, DMDD significantly increased by 0.62 standard deviations (*p*-value < 0.001). This model provided a good fit for the data, as suggested by the non-significant chi-square (*p* = 0.571) and seven other goodness-of-fit indices (SRMR = 0.014, RMSEA < 0.001, GFI = 0.998, IFI = 1.006, CFI = 1.000, NFI = 0.997, and TLI = 1.0383).

The mediation models, as depicted in Figure 2, evaluated the strength of the indirect relationship while controlling for the direct effect of ADHD on DMDD. The direct path from ADHD to DMDD remained significant (*p* = 0.001). In addition, the standardized indirect (mediated) effect of ADHD on DMDD was 0.044 (=0.21 × 0.21). That is, due to the indirect (mediated) effect of ADHD on DMDD, when ADHD increases by 1 standard deviation, DMDD increases by 0.044 standard deviations. The GFIs of this mediation model provided an excellent fit for the data (chi-square = 1.087, *p* = 0.297, SRMR = 0.026, RMSEA = 0.029, GFI = 0.996, IFI = 0.999, CFI = 0.999, NFI = 0.992, and TLI = 0.993). Notably, the standardized direct effect of CIAS on DMDD was 0.21 (*p* = 0.005) after adjusting for the direct effect of ADHD on DMDD.

## 4. Discussion

The concept of IGD mediating ADHD pathways leading to DMDD has not been entirely clear before. In this study, we demonstrated how gaming disorders drive ADHD to DMDD. Gaming plays a mediating role to escalate the effects of ADHD to DMDD. Under the hypothesis, the SEM analysis (analysis of symptom-development pathways) found gaming disorders worsen the symptoms of ADHD to DMDD. IGD was a risk factor and was associated with emotional dysregulation among ADHD youths.

If we explain this finding using the research-domain-criteria-dimensions-model perspective, children with ADHD have a deficit in the domain of cognition (specifically in working memory) and positive valence (in rewarding anticipation/delay/receipt) [35]. Children with IGD may exhibit problems in the domains of negative valence systems, positive valence systems, cognitive systems, social process systems, and arousal and regulatory systems [36]. Therefore, IGD and ADHD may have mixed or overlapped disturbances in the domains of executive function, incentive salience, and negative emotionality [37]. Our results indicating that gaming disorder might aggravate the negative emotional symptoms of ADHD leading to emotional dysregulation is congruous with the research domain criteria model perspective. This SEM pathway analysis indicates that IGD may indeed worsen the symptoms of inattention, hyperactivity/impulsivity, and ODD in children with ADHD.

We explain below a vicious cycle that illustrates why IGD plays a mediating role and worsens the symptoms of ADHD, even developing negative moods. A vicious cycle starts with gaming-addicted ADHD youths characterized in our descriptive analysis: these youths that are more likely to have poor interpersonal relationships are clinically more comorbid with DMDD, have older parents, and have parents with more marital discord and a poorer parenting strategy for managing stress compared with ADHD youth without IGD. This implies that such youths live in a vulnerable state with their severe symptoms of ADHD and emotional irritability, coupled with poor interpersonal relationships. Through the process of long-term addiction to gaming, these vulnerable youths become affected by DMDD. The cycle becomes vicious, as IGD might lead ADHD youth to spend more time gaming to avoid family or social interactions; gradually, gaming addiction leads them to become lonelier and more irritable, especially when their excessive gaming behavior is curtailed.

One study named “The association between internet addiction and psychiatric co-morbidity: a meta-analysis” by Roger C Ho et al. from Singapore in 2014 [38] found the association between internet addiction (IA) and alcohol abuse, ADHD, and depression. DMDD is a new depression-related mental disorder. In line with their finding, we found similar links between IA and other mood spectrum or ADD spectrum disorders. In addition, recently, mental health experts indeed found that the involvement of the serotonin genotype in IA and depression suggests that mood spectrum or ADD spectrum disorders may share similar neurochemical changes. This study aims to explore new DMDD-like symptoms noticed in untreated ADHD youth associated with internet addiction. Such a study result might remind child and adolescent mental health experts to keep more eyes on these ADHD children especially if they are also addicted to gaming.

Our findings detail the etiology from the genetic and environmental aspects regarding the development of gaming disorder. For youths with IGD, the untreated ADHD was genetic loading that led youths with IGD to exhibit severe symptoms of ADHD, impulsivity, and irritability. Gradually, IGD might enhance the genetic risk of untreated ADHD youths further, presenting more severe symptoms similar to DMDD. In addition, the environmental and family risks include untreated ADHD, living in an environment of low family cohesion, family conflicts, and poor family relationships and family functioning [39]. Through the process of long-term addiction to gaming, these untreated ADHD children become more irritable, even disruptive, in their moods. Thus, for treating families that have an internet-gaming-addicted ADHD youth with an irritable mood, the development of a biopsychosocial model through recent neuropsychiatric expertise is strongly needed. This implies combining pharmacotherapy for ADHD and/or antipsychotic drugs for disruptive mood with a parental program, which is especially needed. The parental program for these gaming-addicted, emotional youth should include cognitive behavior therapy, parents’ marital therapy, improving communication with gaming-addicted youth, and parental stress management. Additionally, principles of healthy digital use are essential treatment interventions for these ADHD youth with mood dysregulation.

In the last two decades, more scholars have focused on other comorbid psychiatric disorders among gaming-addicted adolescents, such as IGD co-occurring more with depression [40], social anxiety, nicotine use disorder, alcohol use disorder, other substance use disorders [41], somatoform disorders, pathological gambling, adult-type ADHD symptoms, sleep disturbances, suicidal ideation, suicidal plans [42], social phobia [43], phobias, psychosis except for paranoia [44], loneliness and problematic behavior disinhibiting [45], and withdrawal psychosis [46,47]. However, this study is the first to find that children with ADHD present increasing irritability, anger, and poor tempers, and their symptoms appear similar to DMDD. This severe irritable mood characteristic is closely intensified by the long-term process of excessive gaming on youth with untreated ADHD, Oppositional Defiant Disorder, or Conduct Disorder.

This study has the following limitations. First, in this study, DMDD was diagnosed by a psychiatrist according to the new criteria in the DSM-V; however, the stability of the DMDD diagnosis after the gaming disorder was not followed up after this study. Additionally, the diagnosis of DMDD needs to fulfill the criteria of unreasonable mood dysregulation and the age at onset being before 10 years. Thus, we regarded our subjects having DMDD-like symptoms because they had disruptive mood dysregulation recently, but we had no history of mood dysregulation starting before 10 years old. This is why we used the term “DMDD-like symptoms”. Therefore, the differentiation between the real DMDD and withdrawal symptoms of gaming disorders resembling DMDD symptoms need to be considered. In addition, DMDD is a disease without any questionnaire or measurement at the time of this study. We used a Likert scale, 0 to 3, to express the symptom severity of the DMDD criteria of the DSM-5. In the future, a more validated DMDD questionnaire is crucially needed to study more about DMDD and withdrawal symptoms of internet gaming disorder on children and adolescents. Second, for convenience, only children and adolescents with ADHD diagnostic antecedents were selected as risks. Other risks, such as socially accepted internet overusing behavior leading to both parents and children being IGD victims, may also lead ADHD children with IGD to develop DMDD-related symptoms. One recent study named “What Factors Are Most Closely Associated With Mood Disorders in Adolescents During the COVID-19 Pandemic? A Cross-Sectional Study Based on 1771 Adolescents in Shandong Province, China” by Ziyuan Ren from China in 2021 [48] found that the occurrence of symptoms of anxiety and depression were 28.3 and 30.8% among the participants and poor sleep quality was the most significant risk factor for mood disorders among Chinese adolescents. Indeed, we did notice poor sleep quality among these ADHD youth too. Only because this is a SEM analytic study to find the developmental pathway from ADHD to DMDD-like symptoms, we did not set a variable for poor sleep quality data for these emotionally dysregulated youth. A future study may consider the risk factor of poor sleep in the study of children and adolescents with ADHD comorbid with IGD. Despite these limitations, the application of SEM to explore the multiple correlated risks leading to juvenile mood dysregulation and the fits all appear good or appropriate, indicating SEM is a useful technique to elucidate the simultaneous risks leading to more severe mental disorders.

Our future society is likely to contain more youths with IGD [49]. Child psychiatrists should recognize and be cautious of the silent hazard triggered by gaming disorder, especially for youths with untreated ADHD. For internet-gaming-addicted youths suffering severely from ADHD and exhibiting warning signs of DMDD, such as irritable mood and aggressive behavior, we suggest an intensive treatment program that combines pharmacotherapy for ADHD and/or antipsychotics pharmacotherapy for children with disruptive mood and cognitive behavior therapy for youths with IGD and their parents. Before the COVID-19 pandemic, certain countries may have required greater attention to the harmful consequences of internet addiction for adolescents. During COVID-19 pandemic, more school students were locked down at home and indeed became students with IGD gradually. Simultaneously, more youths were recognized as having untreated ADHD, DMDD, or depression, which are highly associated with IGD. This study’s results emphasize that after the COVID-19 pandemic, there will be more children and adolescents who become internet-addicted due to their excessive use during the COVID-19 pandemic. Therefore, it is necessary to develop more prevention and treatment strategies soon as attempts are made to attain a new normal.

In summary, ADHD should be treated early to prevent serious consequences such as antisocial personality disorder and substance-related and addictive disorders [50]. IGD among youths with ADHD is neglected and remains undertreated, but it is a new mental disorder in our society. The findings of this study indicated that gaming disorder indirectly mediates ADHD in children and presents irritable symptoms similar to DMDD. Therefore, children with ADHD should no longer be neglected or undertreated, especially in some developing countries. Child and adolescent psychiatrists and pediatric-related ADHD experts should regard excessive gaming behavior among children and adolescents not only a game-playing problem but also a serious risk that can lead children with ADHD to have DMDD-like symptoms. In addition, we should consider IGD as a warning sign of possible neurodevelopmental disorder escalation of ADHD to disruptive mood dysregulation symptoms among children and adolescents.

## 5. Conclusions

The findings suggest that IGD mediates ADHD’s progression to disruptive mood dysregulation. Intensive biopsychosocial interventions are warranted for ADHD youths with IGD. More children and adolescents became mood-dysregulated after excessive gaming during the COVID-19 pandemic; this study’s results suggest that child mental health experts develop earlier detection and prevention strategies for children and adolescents hidden behind internet addiction.

## Figures and Tables

**Figure 1 ijerph-19-06648-f001:**
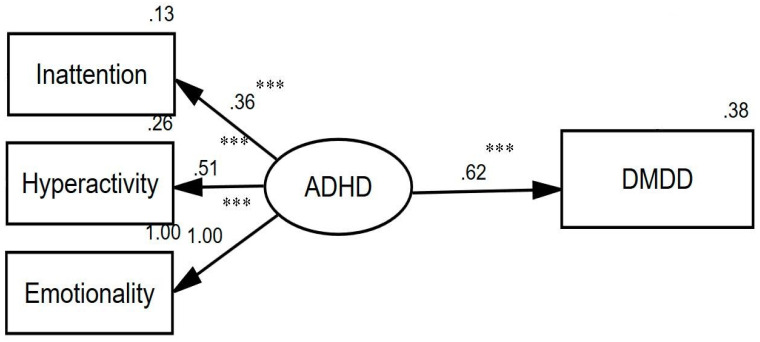
The basic model depicting the direct relationship between ADHD and DMDD. Circles represent unobserved latent variables. Rectangles represent observed measured variables. Values are standardized path coefficients. Goodness-of-fit indicators: Chi-square = 0.322 (*p* = 0.571), SRMR = 0.014, RMSEA < 0.001, GFI = 0.998, IFI = 1.006, CFI = 1.000, NFI = 0.997, and TLI = 1.0383. *** *p* < 0.001. ADHD: attention deficit hyperactivity disorder; DMDD: disruptive mood dysregulation disorder; SRMR = standardized root-mean-square-residual; RMSEA = root-mean-squared error of approximation; GFI: goodness of fit; IFI: incremental fit index; CFI = comparative fit index; NFI: normed fit index; TLI = Tucker–Lewis index.

**Figure 2 ijerph-19-06648-f002:**
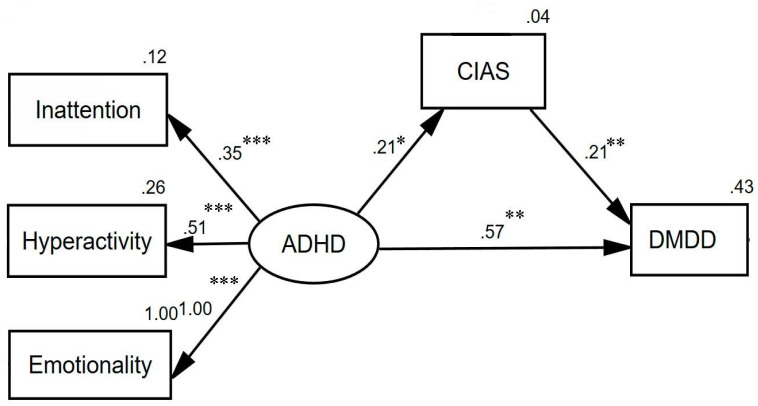
The mediation models. Circles represent unobserved latent variables. Rectangles represent observed measured variables. Values are standardized path coefficients. Goodness-of-fit indicators: Chi-Square = 1.087 (*p* = 0.297), SRMR = 0.026, RMSEA = 0.029, GFI = 0.996, IFI = 0.999, CFI = 0.999, NFI = 0.992, and TLI = 0.993. * *p* < 0.05, ** *p* < 0.01, and *** *p* < 0.001. ADHD: attention-deficit/hyperactivity disorder; DMDD: disruptive mood dysregulation disorder; CIAS: Chen’s Internet Addiction Scale; SRMR: standardized root-mean-square residual; RMSEA: root-mean-squared error of approximation; GFI: goodness of fit; IFI: incremental fit index; CFI: comparative fit index; NFI: normed fit index; TLI: Tucker–Lewis index.

**Table 1 ijerph-19-06648-t001:** Comparisons of the baseline characteristics of the children with ADHD between IGD and non-IGD groups.

		Internet Addiction (CIAS ≥ 57)		*p*-Value
		No (*n* = 49)	Yes (*n* = 53)	
Gender	Male	38 (77.6%)	32 (60.4%)	0.087 ^a^
	Female	11 (22.4%)	21 (39.6%)	
School performance	Average	24 (50.0%)	23 (44.2%)	0.689 ^a^
	Bad	24 (50.0%)	29 (45.8%)	
Interpersonal relationships	Good	36 (75.0%)	25 (48.1%)	0.008 ^a^
	Bad	12 (25.0%)	27 (51.9%)	
Comorbid diagnoses				
ODD	Yes	34 (69.4%)	45 (84.9%)	0.096 ^a^
	No	15 (30.6%)	8 (15.1%)	
CD	Yes	0 (0%)	2 (3.8%)	0.496 ^a^
	No	49 (100.0%)	51 (96.2%)	
DMDD-like	Yes	26 (53.1%)	42 (79.2%)	0.006 ^a^
	No	23 (46.9%)	11 (20.8%)	
Anxiety	Yes	0 (0.0%)	1 (1.9%)	1.000 ^a^
	No	49 (100.0%)	52 (98.1%)	
Adjustment disorder	Yes	0	0	
	No	49 (48.0%)	53 (52.0%)	
Somatic symptom disorder	Yes	3 (6.1%)	3 (5.7%)	1.000 ^a^
	No	46 (93.9%)	50 (94.3%)	
Tics	Yes	5 (10.2%)	3 (5.7%)	0.476 ^a^
	No	44 (89.8%)	50 (94.3%)	
Tourette’s syndrome	Yes	3 (6.1%)	4 (7.5%)	1.000 ^a^
	No	46 (93.9%)	49 (92.5%)	
Speech sound disorder	Yes	0 (0.0%)	1 (1.9%)	1.000 ^a^
	No	49 (100.0%)	52 (98.1%)	
Language disorder history	Yes	1 (2.0%)	1 (1.9%)	1.000 ^a^
	No	48 (98.0%)	52 (98.1%)	
Internet gaming	Yes	18 (36.7%)	49 (92.5%)	<0.001 ^a^
disorder	No	31 (63.3%)	4 (7.5%)	
Depression	Yes	0 (0.0%)	1 (1.9%)	1.000 ^a^
	No	49 (100.0%)	52 (98.1%)	
Subtype	Combined	35 (71.4%)	30 (56.6%)	0.150 ^a^
	Inattentive	14 (28.6%)	23 (43.4%)	
Family hereditary	Yes	11 (22.4%)	10 (18.9%)	0.807 ^a^
history	No	38 (77.6%)	43 (81.1%)	
Sibling suffering from	Yes	11 (22.4%)	9 (17.0%)	0.619 ^a^
ADHD	No	38 (77.6%)	44 (83.0%)	
Parents suffering from	Yes	13 (26.5%)	19 (35.8%)	0.394 ^a^
ADHD in Childhood	No	36 (73.5%)	34 (64.2%)	
Strategy of parents to	Appropriate	31 (64.6%)	23 (43.4%)	0.046 ^a^
deal with stress	Inappropriate	17 (35.4%)	30 (56.6%)	
Parental understanding	Yes	21 (42.9%)	21 (39.6%)	0.841 ^a^
of ADHD	No	28 (57.1%)	32 (60.4%)	
Parental marital	Satisfied	43 (87.8%)	38 (71.7%)	0.053 ^a^
satisfaction	Unsatisfied	6 (12.2%)	15 (28.3%)	
Working days online	≥1 h	23 (46.9%)	43 (81.1%)	<0.001 ^a^
Chat or play game	<1 h	26 (53.1%)	10 (18.9%)	
Holiday online chat or	≥3 h	21 (42.9%)	45 (84.9%)	<0.001 ^a^
play game	<3 h	28 (57.1%)	8 (15.1%)	
Drug response	Good	14 (50.0%)	11 (31.4%)	0.195 ^a^
	Bad	14 (50.0%)	24 (68.6%)	
Parenting group therapy	Yes	7 (23.3%)	8 (20.0%)	0.775 ^a^
	No	23 (76.7%)	32 (80.0%)	
Compliance	Good	13 (48.1%)	10 (27.8%)	0.118 ^a^
	Bad	14 (51.9%)	26 (72.2%)	
Age		10.16 ± 3.05	12.29 ± 3.69	0.002 ^b^
Height		138.80 ± 18.15	148.98 ± 18.71	0.007 ^b^
Weight		35.89 ± 15.06	45.85 ± 18.24	0.003 ^b^
Age of father		42.63 ± 6.30	46.76 ± 7.87	0.005 ^b^
Age of mother		40.22 ± 7.25	43.53 ± 6.98	0.021 ^b^
No. of Comorbidity		1.90 ± 1.21	2.89 ± 0.91	<0.001 ^c^

IGD: Internet Gaming Disorder; ^a^: Fisher’s Exact test; ^b^: Independent *t*-test; ^c^: Mann–Whitney U test.

**Table 2 ijerph-19-06648-t002:** Zero-order correlations among study measures.

	Inattention	Hyperactivity	Emotionality	CIAS	DMDD
Inattention	1	0.476 ***	0.355 ***	0.270 **	0.177
Hyperactivity		1	0.508 ***	0.020	0.141
Emotionality			1	0.211 *	0.616 ***
CIAS				1	0.350 ***
DMDD					1

CIAS: Chen Internet Addiction Scale; DMDD: Disruptive Mood Dysregulation Disorder; *: *p* < 0.05; **: *p* < 0.01; ***: *p* < 0.001.

## Data Availability

Data openly available in a public repository.

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
