# Peer review of "Structural Equation Modeling (SEM): Gaming Disorder Leading Untreated Attention-Deficit/Hyperactivity Disorder to Disruptive Mood Dysregulation"

_ijerph, 2022, doi:10.3390/ijerph19116648_

Round 1
Reviewer 1 Report
Review Report:
In this manuscript, the authors have highlighted that Internet gaming disorder (IGD) in youths likely leads to disruptive mood dysregulation, especially among those with attention deficit hyperactivity disorder (ADHD). However, whether IGD mediates the pathways leading ADHD to disruptive emotional dysfunction remains unclear. Their work aims to elucidate the direct or indirect influence of IGD on ADHD. They have used the methodology of the Swanson, Nolan, and Pelham, Version IV questionnaire to evaluate symptoms of ADHD and oppositional defiant disorder, and the Chen gaming disorder scale to measure IGD. A psychiatrist was recruited to diagnose ADHD, IGD, and disruptive mood dysregulation disorder (DMDD) - like symptoms. The authors have additionally applied structural equation modeling to evaluate the role of IGD in mediating ADHD to disruptive mood dysregulation. Their results displayed that amongst a total of 102 ADHD youths, 53 (52%) of them were diagnosed with IGD, were significantly more likely to have poor interpersonal relationships (p < 0.01) and DMDD-like symptoms (p < 0.01) than ADHD youths without IGD. They concluded that IGD played a mediating role in increasing the risk of disruptive mood dysregulation in ADHD youths. Their findings suggest that IGD mediates ADHD to disruptive mood dysregulation and intensive biopsychosocial interventions are warranted for ADHD youths with IGD.
Overall Recommendation: Accept after Minor Revisions: The paper is in principle accepted after revision based on the reviewer’s comments.
Review Report:
However, there are a few minor points that need to be addressed:
- The abstract should highlight the relevance of this research in the last few lines. The introduction should have one last paragraph stating why these study results are important and valid for the aims and scope of the International Journal of Environmental Research and Public Health, considering it is a peer-reviewed scientific journal that publishes original articles, critical reviews, research notes, and short communications in the interdisciplinary area of environmental health sciences and public health. The authors need to highlight the application of this research and why their findings are relevant in the context of public health. In the current stage, the abstract appears as only a summary of their results.
- The titles of every table should follow with few lines describing the content and the relevance of the tables.
- The figures are very simplistic and need some more detailed work in terms of the explanation, as to what the values on the arrows represent. It is a bit confusing and needs a simplistic representation with detailed explanation of each individual structure, values, methodology and statistical power.
Author Response
Review 1: Comments and Suggestions for Authors
In this manuscript, the authors have highlighted that Internet gaming disorder (IGD) in youths likely leads to disruptive mood dysregulation, especially among those with attention deficit hyperactivity disorder (ADHD). However, whether IGD mediates the pathways leading ADHD to disruptive emotional dysfunction remains unclear. Their work aims to elucidate the direct or indirect influence of IGD on ADHD. They have used the methodology of the Swanson, Nolan, and Pelham, Version IV questionnaire to evaluate symptoms of ADHD and oppositional defiant disorder, and the Chen gaming disorder scale to measure IGD. A psychiatrist was recruited to diagnose ADHD, IGD, and disruptive mood dysregulation disorder (DMDD) - like symptoms. The authors have additionally applied structural equation modeling to evaluate the role of IGD in mediating ADHD to disruptive mood dysregulation. Their results displayed that amongst a total of 102 ADHD youths, 53 (52%) of them were diagnosed with IGD, were significantly more likely to have poor interpersonal relationships (p < 0.01) and DMDD-like symptoms (p < 0.01) than ADHD youths without IGD. They concluded that IGD played a mediating role in increasing the risk of disruptive mood dysregulation in ADHD youths. Their findings suggest that IGD mediates ADHD to disruptive mood dysregulation and intensive biopsychosocial interventions are warranted for ADHD youths with IGD.
Overall Recommendation: Accept after Minor Revisions: The paper is in principle accepted after revision based on the reviewer’s comments.
Ans: thank you for your acceptance. We will revise this article based on the reviewer’s comments soon.
Review Report:
However, there are a few minor points that need to be addressed:
The abstract should highlight the relevance of this research in the last few lines. The introduction should have one last paragraph stating why these study results are important and valid for the aims and scope of the International Journal of Environmental Research and Public Health, considering it is a peer-reviewed scientific journal that publishes original articles, critical reviews, research notes, and short communications in the interdisciplinary area of environmental health sciences and public health. The authors need to highlight the application of this research and why their findings are relevant in the context of public health. In the current stage, the abstract appears as only a summary of their results.
Ans: Thank you for good suggestion. We will add the following word to apply this research and why their findings are relevant of public mental health in the last few lines of abstract and last paragraph of introduction.
“For early preventing more children and adolescent becoming mood dysregulated after excessive gaming during Covid-19 pandemic, this study result suggested the child mental health expertise to develop earlier detection and prevention strategy for children and adolescents hidden behind internet addiction”.
The titles of every table should follow with few lines describing the content and the relevance of the tables.
Ans: Thank you for good suggestion. we have corrected every table.
The figures are very simplistic and need some more detailed work in terms of the explanation, as to what the values on the arrows represent. It is a bit confusing and needs a simplistic representation with detailed explanation of each individual structure, values, methodology and statistical power.
Ans:thank you again. For the sake of more detailed work in terms of the explanation, we have removed the CFA part (in statistical method section, since we had presented that). We also revised the results section and both Figures to explain it with more detail.

Reviewer 2 Report
1) As the authors mentioned that this paper is related to the COVID-19 pandemic and submitted to the theme of the COVID-19 pandemic, the author should mention the number of COVID-19 cases and death in Taiwan and the policy on school attendance during the period of study.
2) Under the methods, the authors should state the recruitment period and how severe the COVID-19 situation in Taiwan was during the study period.
3) Under the method, I hope the authors can discuss the following:
A study on Pubmed shows "Our analyses demonstrated a significant and positive association between IA and alcohol abuse (OR = 3.05, 95% CI = 2.14-4.37, z = 6.12, P < 0.001), attention deficit and hyperactivity (OR = 2.85, 95% CI = 2.15-3.77, z = 7.27, P < 0.001), depression (OR = 2.77, 95% CI = 2.04-3.75, z = 6.55, P < 0.001)" (PMID: 24947851)
It seems that it may not be just gaming. Internet addiction is associated with ADHD and MDD in adults before the pandemic. Can the authors discuss how their findings is relevant for the future (i.e. adulthood and when the COVID-19 pandemic is normalized, all people living with COVID and life as usual).
4) This study too focused on Internet gaming addiction, ADHD, and MDD but did not explore important factors to adolescents during the COVID-19 pandemic. I came across this study on Pubmed
"We found that the poor sleep quality was the most significant risk factor for mood disorders among Chinese adolescents." (PMID: 34603106)
This study did not report the sleep quality of participants and please list this as a limitation.
I recommend the authors to discuss whether the intervention proposed by the above study "sleep quality-resilience-coping strategy-social support-perceived social status" is relevant to Taiwanese adolescents.
Author Response
Review 2: Comments and Suggestions for Authors
As the authors mentioned that this paper is related to the COVID-19 pandemic and submitted to the theme of the COVID-19 pandemic, the author should mention the number of COVID-19 cases and death in Taiwan and the policy on school attendance during the period of study.
Ans: According to data of Taiwan Center of Disease Control, the number of COVID-19 cases is 397,504 from beginning to 2022/05/10 and death in Taiwan is 391. When we compared to other country, the case number is few and there is no specific school attendance restriction during the period of study. This study is focusing more on the influence of IGD on ADHD children and adolescent. We did not find out related reference on this topic.
Under the methods, the authors should state the recruitment period and how severe the COVID-19 situation in Taiwan was during the study period.
Ans: Thank you for suggestion. “This study was recruited on the beginning of COVID-19 situation in Taiwan. But this is a special statistical analysis called Structural equation modeling (SEM) applied to evaluate the role of IGD in mediating ADHD to disruptive mood dysregulation. We have no data of how severe the COVID-19 situation in Taiwan was during the study period influence this study result.”
We have added this part under the method part, on page 5 , line 118-122 .
Under the method, I hope the authors can discuss the following: A study on Pubmed shows "Our analyses demonstrated a significant and positive association between IA and alcohol abuse (OR = 3.05, 95% CI = 2.14-4.37, z = 6.12, P < 0.001), attention deficit and hyperactivity (OR = 2.85, 95% CI = 2.15-3.77, z = 7.27, P < 0.001), depression (OR = 2.77, 95% CI = 2.04-3.75, z = 6.55, P < 0.001)" (PMID: 24947851)
Ans: thank you for suggestion. This is study named “The association between internet addiction and psychiatric co-morbidity: a meta-analysis” by Roger C Ho et al. from Singapore on 2014. They found the association between internet addiction (IA) to alcohol abuse, ADHD, depression. DMDD is a new depression related mental disorder. In line with their finding, we found similar link between IA and other mood spectrum or ADD spectrum disorder. In addition, recently mental health expertise indeed found the involvement of the serotonin genotype in IA and depression suggests that mood spectrum or ADD spectrum disorder may share similar neurochemical changes. This study aims on new DMDD like symptom noticed on untreated ADHD youth associated with internet addiction. Such study result might remind child and adolescent mental health expertise to keep more eyes on these ADHD children especially while youth also was addicted by gaming.
We have added this part in the discussion part on page 11 , line 267-277.
It seems that it may not be just gaming. Internet addiction is associated with ADHD and MDD in adults before the pandemic. Can the authors discuss how their findings is relevant for the future (i.e., adulthood and when the COVID-19 pandemic is normalized, all people living with COVID and life as usual).
Ans: thank you for suggestion. Before the COVID-19 pandemic, some countries may require greater attention to the harmful consequences of Internet addiction for adolescents. During COVID-19 pandemic, more school student locked home and indeed became student with IGD gradually. Simultaneously, more youth noticed to have untreated ADHD, DMDD, or depression which is highly associated with IGD. This study results emphasize that after the COVID-19 pandemic, there will be more children and adolescents who become Internet addiction due to their excessive use during the COVID-19 pandemic. Therefore, it is necessary to develop more prevention and treatment strategies soon when the COVID-19 pandemic is normalized or all people living with COVID and life as usual.
We have added this part on discussion on page 14 , line 344 to 353.
4) This study too focused on Internet gaming addiction, ADHD, and MDD but did not explore important factors to adolescents during the COVID-19 pandemic. I came across this study on Pubmed"We found that the poor sleep quality was the most significant risk factor for mood disorders among Chinese adolescents." (PMID: 34603106) This study did not report the sleep quality of participants and please list this as a limitation.
Ans: thank you for good suggestion. This study named” What Factors Are Most Closely Associated With Mood Disorders in Adolescents During the COVID-19 Pandemic? A Cross-Sectional Study Based on 1,771 Adolescents in Shandong Province, China” by Ziyuan Ren from China on 2021. They found the occurrence of symptoms of anxiety and depression were 28.3 and 30.8% among the participants and poor sleep quality was the most significant risk factor for mood disorders among Chinese adolescents. Indeed, we did notice poor sleep quality among these ADHD youth too. Only because this is SEM analytic study to find the developmental pathway from ADHD to DMDD like symptom. Therefore, we did not set variable of poor sleep quality data for these emotional dysregulated youth. Future study may consider the risk factor of poor sleep while we study the children and adolescent children with ADHD comorbid with IGD.
We have listed this point on limitation part, as following on page 12 , line 324 to 333.
I recommend the authors to discuss whether the intervention proposed by the above study "sleep quality-resilience-coping strategy-social support-perceived social status" is relevant to Taiwanese adolescents.
Ans: Thank you for suggestion. the above study provided "sleep quality-resilience-coping strategy-social support-perceived social status" as following:
- The first step is to emphasize the significance of healthy sleep and to teach students how to get good sleep and deal with sleep disorders (sleep quality).
- The second step is to foster adolescents' ability to bounce back with new learning and strength when facing adversity (resilience).
- The third step is to share with students strategies for the management of uncomfortable emotions by using healthy coping skills and avoiding unhealthy coping skills (coping strategies).
- The fourth step is to encourage students to build authentic relationships with their families, friends, teachers, and others (social support).
- The last step is to help students discover their core values and encourage students to ask for help when they encounter unfair treatment (social/school status).
Inside our article, we have suggested the following intervention of IGD on discussion according to the study result by SEM.
“Thus, for treating a family that has Internet gaming addicted ADHD youth with irritable mood, developing a biopsychosocial model by recent neuropsychiatric expertise is strongly needed. This implies combining pharmacotherapy for ADHD and/or antipsychotic drugs for disruptive mood with a parental program is essentially needed. The parental program for these gaming addicted emotional youth should include cognitive behavior therapy, parent’s marital therapy, improving communication with gaming addicted youth, and parental stress management. Also, principle of healthy digital using is essentially important treatment intervention for these ADHD youth with mood dysregulation.”

Reviewer 3 Report
Dear authors,
I detected some problems with your paper. Some of them are small, and some of them are crucial.
-No definition of diagnoses
There are several problems with the definition of diagnoses. For e.g:
You are focusing on ADHD, IGD, DMDD, and other diagnoses (The exact list is on page 12). However, you never described/detailed these diagnoses. As you see, you mentioned them on pages 1 and 2 first and second paragraphs but there is no proper diagnosis explanation. Simply, "What kind of mental disorders are they? How affect human life?" This part is quite important and it has to be clear because the paper has to be understandable for other scientists too, not just psychiatrists or psychologists.
-Scale problems
You mentioned you used CIAS (https://psycnet.apa.org/record/2004-10292-005) But they wrote the scale name as "Chen gaming disorder scale (CIA)". In some paragraphs, you mentioned CIAS sometimes it was CIA. Which name is true? In the meantime, there is a game version of that scale (https://pubmed.ncbi.nlm.nih.gov/31661785/) Did you use that one or the original one? Also, you mentioned it is a game scale but you did not describe it as a game version. Is it CIAS-G? This part is competently messy, there is no proper explanation, Abbreviation is not correct, because the reference says it is CIAS or CIAS-R.
Simply, they have to re-write that part and explain it better. You did not write Cronbach alpha. You need to put proper references for that part.
The authors did not explain properly, how did measure DMDD-like symptoms?
-Organization problems
Why are descriptions of scales inside of the "Baseline characteristics" part?
-Why some of the parts of your article are exactly the same as in your previous article?
This is a problematic topic because you published in 2020, "Treatment Efficacy of Internet Gaming Disorder With Attention Deficit Hyperactivity Disorder and Emotional Dysregulaton" https://doi.org/10.1093/ijnp/pyaa010
You can find the link to these paragraphs:
https://i.ibb.co/cxSDSjB/p1.jpg
Author Response
Review 3: Comments and Suggestions for Authors
Dear authors,I detected some problems with your paper. Some of them are small, and some of them are crucial.
-No definition of diagnoses
There are several problems with the definition of diagnoses. For e.g:
You are focusing on ADHD, IGD, DMDD, and other diagnoses (The exact list is on page 12). However, you never described/detailed these diagnoses. As you see, you mentioned them on pages 1 and 2 first and second paragraphs but there is no proper diagnosis explanation. Simply, "What kind of mental disorders are they? How affect human life?" This part is quite important and it has to be clear because the paper has to be understandable for other scientists too, not just psychiatrists or psychologists.
Ans: ADHD is a neurodevelopmental disorder, commonly noticed among child and adolescent with prevalence of 9%. IGD is new formal mental disorder and also very commonly seen among recent among child and adolescent with prevalence ranging from 4-6% (in European countries) to 13.5% (in China). This study result highlight if untreated ADHD live in the internet addicted life, they will become mood disrupted state instead of depression. Depression is a mood disorder commonly noticed among adult instead of children and adolescent. Here we suggest children mental health expertise face recent youth with IGD properly and provide early IGD preventive intervention to prevent untreated ADHD become DMDD symptom which eventually bring bad influence on their personality development.
We have added this part on discussion part on page 13 , line 354-362.
-Scale problems
You mentioned you used CIAS (https://psycnet.apa.org/record/2004-10292-005) But they wrote the scale name as "Chen gaming disorder scale (CIA)". In some paragraphs, you mentioned CIAS sometimes it was CIA. Which name is true? In the meantime, there is a game version of that scale (https://pubmed.ncbi.nlm.nih.gov/31661785/) Did you use that one or the original one? Also, you mentioned it is a game scale but you did not describe it as a game version. Is it CIAS-G? This part is competently messy, there is no proper explanation, Abbreviation is not correct, because the reference says it is CIAS or CIAS-R.
Ans: CIAS is Chen internet addiction scale. We have corrected the writing as consistent ways as CIAS.
Simply, they have to re-write that part and explain it better. You did not write Cronbach alpha. You need to put proper references for that part.
Ans: thank you for suggestion. CIAS has good internal consistency of the scale, with Cronbach's alpha values between 0.79 and 0.93 for the subscales. Inside this article, we have explained as “The internal reliability of the scale and the subscales in the original study ranged from 0.79 to 0.93”. But now we will explain it as “CIAS has good internal consistency of the scale, with Cronbach's alpha values between 0.79 and 0.93 for the subscales”.
We have rewritten this part on page 7, line 146 to 149.
The authors did not explain properly, how did measure DMDD-like symptoms?
Ans: Because the diagnosis of DMDD needs to fulfill the criteria of unreasonable mood dysregulation and the age at onset is before 10 years. Thus, we regarded our subjects having DMDD-like symptoms because they had only disruptive mood dysregulation recently, but we had no history of mood dysregulation starting before 10 years old. This is why we only use the terms of DMDD-like symptoms.
We have further explained this on limitation part again on page 13 , line 310 to 314.
-Organization problems
Why are descriptions of scales inside of the "Baseline characteristics" part?
Ans: we have separated them.
-Why some of the parts of your article are exactly the same as in your previous article?
This is a problematic topic because you published in 2020, "Treatment Efficacy of Internet Gaming Disorder With Attention Deficit Hyperactivity Disorder and Emotional Dysregulaton" https://doi.org/10.1093/ijnp/pyaa010 .You can find the link to these paragraphs:https://i.ibb.co/cxSDSjB/p1.jpg
Ans: thank you for requiring. Indeed, we used the data of IRB No: 19MMHIS387e, titled “Treatment Effect of Combining Pharmacotherapy and Cognitive Behavior Therapy on Child and Adolescent with Internet Gaming Disorder”. Patients were recruited from the Out-Patient Units of Mackay Memorial Hospital in Taipei, Taiwan, and the research protocol was approved by the hospital’s institutional review board. Written informed consent was obtained from each patient consistent with the institutional review board guidelines. Only we used special SEM statistical analysis to figure out the link between ADHD, IGD and DMDD like symptom in this SEM study.

Round 2
Reviewer 3 Report
Review 3: Comments and Suggestions for Authors
Dear authors,I detected some problems with your paper. Some of them are small, and some of them are crucial.
-No definition of diagnoses
There are several problems with the definition of diagnoses. For e.g:
You are focusing on ADHD, IGD, DMDD, and other diagnoses (The exact list is on page 12). However, you never described/detailed these diagnoses. As you see, you mentioned them on pages 1 and 2 first and second paragraphs but there is no proper diagnosis explanation. Simply, "What kind of mental disorders are they? How affect human life?" This part is quite important and it has to be clear because the paper has to be understandable for other scientists too, not just psychiatrists or psychologists.
Ans: ADHD is a neurodevelopmental disorder, commonly noticed among child and adolescent with prevalence of 9%. IGD is new formal mental disorder and also very commonly seen among recent among child and adolescent with prevalence ranging from 4-6% (in European countries) to 13.5% (in China). This study result highlight if untreated ADHD live in the internet addicted life, they will become mood disrupted state instead of depression. Depression is a mood disorder commonly noticed among adult instead of children and adolescent. Here we suggest children mental health expertise face recent youth with IGD properly and provide early IGD preventive intervention to prevent untreated ADHD become DMDD symptom which eventually bring bad influence on their personality development.
We have added this part on discussion part on page 13 , line 354-362.
Re3. Feedback: Dear author, you started to mention ADHD in the second paragraph of the background section (Line 49) and IGD in the first paragraph of the background section (Line 42). In these paragraphs and the following paragraphs, you were giving information for related diagnoses. But you never described, what are these sicknesses? You mentioned you have added that part in the "Discussion" section, but this part must be clearer for the definition and this paragraph must be in the "Background" section. Because the reader must understand the concept of these diagnoses before the article goes deeply.
-Scale problems
You mentioned you used CIAS (https://psycnet.apa.org/record/2004-10292-005) But they wrote the scale name as "Chen gaming disorder scale (CIA)". In some paragraphs, you mentioned CIAS sometimes it was CIA. Which name is true? In the meantime, there is a game version of that scale (https://pubmed.ncbi.nlm.nih.gov/31661785/) Did you use that one or the original one? Also, you mentioned it is a game scale but you did not describe it as a game version. Is it CIAS-G? This part is competently messy, there is no proper explanation, Abbreviation is not correct, because the reference says it is CIAS or CIAS-R.
Ans: CIAS is Chen internet addiction scale. We have corrected the writing as consistent ways as CIAS.
Re3. Feedback: Thank you for your effort.
Simply, they have to re-write that part and explain it better. You did not write Cronbach alpha. You need to put proper references for that part.
Ans: thank you for suggestion. CIAS has good internal consistency of the scale, with Cronbach's alpha values between 0.79 and 0.93 for the subscales. Inside this article, we have explained as “The internal reliability of the scale and the subscales in the original study ranged from 0.79 to 0.93”. But now we will explain it as “CIAS has good internal consistency of the scale, with Cronbach's alpha values between 0.79 and 0.93 for the subscales”.
We have rewritten this part on page 7, line 146 to 149.
Re3. Feedback: Thank you for your effort.
The authors did not explain properly, how did measure DMDD-like symptoms?
Ans: Because the diagnosis of DMDD needs to fulfill the criteria of unreasonable mood dysregulation and the age at onset is before 10 years. Thus, we regarded our subjects having DMDD-like symptoms because they had only disruptive mood dysregulation recently, but we had no history of mood dysregulation starting before 10 years old. This is why we only use the terms of DMDD-like symptoms.
We have further explained this on limitation part again on page 13 , line 310 to 314.
Re3. Feedback: Thank you for your effort.
-Organization problems
Why are descriptions of scales inside of the "Baseline characteristics" part?
Ans: we have separated them.
Re3. Feedback: Thank you for your effort.
-Why some of the parts of your article are exactly the same as in your previous article?
This is a problematic topic because you published in 2020, "Treatment Efficacy of Internet Gaming Disorder With Attention Deficit Hyperactivity Disorder and Emotional Dysregulaton" https://doi.org/10.1093/ijnp/pyaa010 .You can find the link to these paragraphs:https://i.ibb.co/cxSDSjB/p1.jpg
Ans: thank you for requiring. Indeed, we used the data of IRB No: 19MMHIS387e, titled “Treatment Effect of Combining Pharmacotherapy and Cognitive Behavior Therapy on Child and Adolescent with Internet Gaming Disorder”. Patients were recruited from the Out-Patient Units of Mackay Memorial Hospital in Taipei, Taiwan, and the research protocol was approved by the hospital’s institutional review board. Written informed consent was obtained from each patient consistent with the institutional review board guidelines. Only we used special SEM statistical analysis to figure out the link between ADHD, IGD and DMDD like symptom in this SEM study.
Re3. Feedback: Dear author, Explanation of "Swanson, Nolan, and Pelham, Version IV Questionnaire (SNAP-IV)", is still looking too like your old article. Can you re-write that part?
From your old article: The SNAP-IV consists of the following items: inattention, hyperactivity/ impulsivity, and oppositional symptoms. These items reflect the core symptoms of ADHD and Oppositional Defiant Disorder as defined in the DSM-IV. The psychometric properties of SNAP-IV-Chinese in Taiwan showed the intra-class correlation coefficients for the 3 subscales of the Chinese SNAP-IV ranged from 0.59 to 0.72 for the parent form and from 0.60 to 0.84 for the teacher form. All subscales of both the parent and teacher forms showed excellent internal consistency with Cronbach’s α > 0.88
From your current article: The SNAP-IV consists of the following items: inattention, hyperactivity/impulsivity, and oppositional symptoms. These items reflect the core symptoms of ADHD and ODD as defined in DSM-IV. The psychometric properties of SNAP-IV-Chinese in Taiwan revealed that the intra-class correlation coefficients for the three subscales of the Chinese SNAP-IV ranged from 0.59 to 0.72 for the parent form and from 0.60 to 0.84 for the teacher form. All subscales of both the parent and teacher forms displayed excellent internal consistency with Cronbach’s α greater than 0.88 (Liu et al., 2006).
Author Response
Review 3: Comments and Suggestions for Authors
Dear authors,I detected some problems with your paper. Some of them are small, and some of them are crucial.
-No definition of diagnoses
There are several problems with the definition of diagnoses. For e.g:
You are focusing on ADHD, IGD, DMDD, and other diagnoses (The exact list is on page 12). However, you never described/detailed these diagnoses. As you see, you mentioned them on pages 1 and 2 first and second paragraphs but there is no proper diagnosis explanation. Simply, "What kind of mental disorders are they? How affect human life?" This part is quite important and it has to be clear because the paper has to be understandable for other scientists too, not just psychiatrists or psychologists.
Ans: ADHD is a neurodevelopmental disorder, commonly noticed among child and adolescent with prevalence of 9%. IGD is new formal mental disorder and also very commonly seen among recent among child and adolescent with prevalence ranging from 4-6% (in European countries) to 13.5% (in China). This study result highlight if untreated ADHD live in the internet addicted life, they will become mood disrupted state instead of depression. Depression is a mood disorder commonly noticed among adult instead of children and adolescent. Here we suggest children mental health expertise face recent youth with IGD properly and provide early IGD preventive intervention to prevent untreated ADHD become DMDD symptom which eventually bring bad influence on their personality development.
We have added this part on discussion part on page 13 , line 354-362.
Re3. Feedback: Dear author, you started to mention ADHD in the second paragraph of the background section (Line 49) and IGD in the first paragraph of the background section (Line 42). In these paragraphs and the following paragraphs, you were giving information for related diagnoses. But you never described, what are these sicknesses? You mentioned you have added that part in the "Discussion" section, but this part must be clearer for the definition and this paragraph must be in the "Background" section. Because the reader must understand the concept of these diagnoses before the article goes deeply.
Ans: thank you for good suggestion. We have moved this part to "Background" section.
-Scale problems
You mentioned you used CIAS (https://psycnet.apa.org/record/2004-10292-005) But they wrote the scale name as "Chen gaming disorder scale (CIA)". In some paragraphs, you mentioned CIAS sometimes it was CIA. Which name is true? In the meantime, there is a game version of that scale (https://pubmed.ncbi.nlm.nih.gov/31661785/) Did you use that one or the original one? Also, you mentioned it is a game scale but you did not describe it as a game version. Is it CIAS-G? This part is competently messy, there is no proper explanation, Abbreviation is not correct, because the reference says it is CIAS or CIAS-R.
Ans: CIAS is Chen internet addiction scale. We have corrected the writing as consistent ways as CIAS.
Re3. Feedback: Thank you for your effort.
Simply, they have to re-write that part and explain it better. You did not write Cronbach alpha. You need to put proper references for that part.
Ans: thank you for suggestion. CIAS has good internal consistency of the scale, with Cronbach's alpha values between 0.79 and 0.93 for the subscales. Inside this article, we have explained as “The internal reliability of the scale and the subscales in the original study ranged from 0.79 to 0.93”. But now we will explain it as “CIAS has good internal consistency of the scale, with Cronbach's alpha values between 0.79 and 0.93 for the subscales”.
We have rewritten this part on page 7, line 146 to 149.
Re3. Feedback: Thank you for your effort.
The authors did not explain properly, how did measure DMDD-like symptoms?
Ans: Because the diagnosis of DMDD needs to fulfill the criteria of unreasonable mood dysregulation and the age at onset is before 10 years. Thus, we regarded our subjects having DMDD-like symptoms because they had only disruptive mood dysregulation recently, but we had no history of mood dysregulation starting before 10 years old. This is why we only use the terms of DMDD-like symptoms.
We have further explained this on limitation part again on page 13 , line 310 to 314.
Re3. Feedback: Thank you for your effort.
-Organization problems
Why are descriptions of scales inside of the "Baseline characteristics" part?
Ans: we have separated them.
Re3. Feedback: Thank you for your effort.
-Why some of the parts of your article are exactly the same as in your previous article?
This is a problematic topic because you published in 2020, "Treatment Efficacy of Internet Gaming Disorder With Attention Deficit Hyperactivity Disorder and Emotional Dysregulaton" https://doi.org/10.1093/ijnp/pyaa010 .You can find the link to these paragraphs:https://i.ibb.co/cxSDSjB/p1.jpg
Ans: thank you for requiring. Indeed, we used the data of IRB No: 19MMHIS387e, titled “Treatment Effect of Combining Pharmacotherapy and Cognitive Behavior Therapy on Child and Adolescent with Internet Gaming Disorder”. Patients were recruited from the Out-Patient Units of Mackay Memorial Hospital in Taipei, Taiwan, and the research protocol was approved by the hospital’s institutional review board. Written informed consent was obtained from each patient consistent with the institutional review board guidelines. Only we used special SEM statistical analysis to figure out the link between ADHD, IGD and DMDD like symptom in this SEM study.
Re3. Feedback: Dear author, Explanation of "Swanson, Nolan, and Pelham, Version IV Questionnaire (SNAP-IV)", is still looking too like your old article. Can you re-write that part?
From your old article: The SNAP-IV consists of the following items: inattention, hyperactivity/ impulsivity, and oppositional symptoms. These items reflect the core symptoms of ADHD and Oppositional Defiant Disorder as defined in the DSM-IV. The psychometric properties of SNAP-IV-Chinese in Taiwan showed the intra-class correlation coefficients for the 3 subscales of the Chinese SNAP-IV ranged from 0.59 to 0.72 for the parent form and from 0.60 to 0.84 for the teacher form. All subscales of both the parent and teacher forms showed excellent internal consistency with Cronbach’s α > 0.88.
From your current article: The SNAP-IV consists of the following items: inattention, hyperactivity/impulsivity, and oppositional symptoms. These items reflect the core symptoms of ADHD and ODD as defined in DSM-IV. The psychometric properties of SNAP-IV-Chinese in Taiwan revealed that the intra-class correlation coefficients for the three subscales of the Chinese SNAP-IV ranged from 0.59 to 0.72 for the parent form and from 0.60 to 0.84 for the teacher form. All subscales of both the parent and teacher forms displayed excellent internal consistency with Cronbach’s α greater than 0.88 (Liu et al., 2006).
Ans: Thank you for suggestion. We have re-written as following: The SNAP-IV is a widely-used rating scale used to screen for ADHD. The SNAP-IV-26 screens for nine symptoms of ADHD hyperactive-impulsive type, nine symptoms of ADHD inattentive type, and eight symptoms of oppositional defiant disorder as defined in the DSM-IV. The psychometric properties of SNAP-IV-Chinese in Taiwan revealed that the intra-class correlation coefficients for the three subscales of the Chinese SNAP-IV ranged from 0.59 to 0.72 for the parent form and from 0.60 to 0.84 for the teacher form. All subscales of both the parent and teacher forms displayed excellent internal consistency with Cronbach’s α greater than 0.88 (Liu et al., 2006).
